# Morphological and Molecular Bases of Male Infertility: A Closer Look at Sperm Flagellum

**DOI:** 10.3390/genes14020383

**Published:** 2023-02-01

**Authors:** Rute Pereira, Mário Sousa

**Affiliations:** 1Laboratory of Cell Biology, Department of Microscopy, ICBAS-School of Medicine and Biomedical Sciences, University of Porto, 4050-313 Porto, Portugal; 2UMIB-Unit for Multidisciplinary Research in Biomedicine, ITR-Laboratory for Integrative and Translational Research in Population Health, University of Porto, 4050-313 Porto, Portugal

**Keywords:** sperm flagellum, infertility, asthenozoospermia, axoneme, morphology

## Abstract

Infertility is a major health problem worldwide without an effective therapy or cure. It is estimated to affect 8–12% of couples in the reproductive age group, equally affecting both genders. There is no single cause of infertility, and its knowledge is still far from complete, with about 30% of infertile couples having no cause identified (named idiopathic infertility). Among male causes of infertility, asthenozoospermia (i.e., reduced sperm motility) is one of the most observed, being estimated that more than 20% of infertile men have this condition. In recent years, many researchers have focused on possible factors leading to asthenozoospermia, revealing the existence of many cellular and molecular players. So far, more than 4000 genes are thought to be involved in sperm production and as regulators of different aspects of sperm development, maturation, and function, and all can potentially cause male infertility if mutated. In this review, we aim to give a brief overview of the typical sperm flagellum morphology and compile some of the most relevant information regarding the genetic factors involved in male infertility, with a focus on sperm immotility and on genes related to sperm flagellum development, structure, or function.

## 1. Introduction

Infertility is defined as the inability to conceive after at least 12 months of regular, unprotected sexual intercourse [1]. Infertility is a major health problem worldwide, with the global disease burden of infertility increasing since 1990. It is estimated to affect 8–12% of couples in the reproductive age group, equally affecting both genders [2]. The causes of infertility are wide-ranging and still not completely understood, with about 30% of infertile couples having no cause identified (named idiopathic infertility). Infertility is influenced by multiple biological factors, making it highly complex and heterogeneous. Therefore, is controversial whether infertility is a disease, a condition, or a symptom.

Male infertility can be manifested by various semen phenotypes which include azoospermia (complete absence of spermatozoa in the ejaculate), oligozoospermia (reduced number of spermatozoa), necrozoospermia (the presence of dead sperm), and qualitative defects associated with sperm cells, including reduced sperm motility (asthenozoospermia), abnormal sperm morphology (teratozoospermia), or the combination of qualitative and quantitative defects (oligoasthenoteratozoospermia). These anomalies could be caused by several factors, such as other health complications (e.g., varicocele, cystic fibrosis, and obesity), infections, lifestyle choices, environmental factors, and genetic factors [3].

Among the semen phenotypes, asthenozoospermia is one of the most observed; it is estimated that more than 20% of infertile men have this condition [4]. A recent report from China analyzed 38,905 infertile male patients and showed that asthenozoospermia was present in 50.5% of these males [5]. Effective motility is necessary for spermatozoa to transit through the female reproductive tract. Normozoospermic (i.e., normal semen phenotype) individuals display two types of physiological motility: activated motility recovered in the ejaculated semen and hyperactivated motility observed at the site and time of fertilization [6,7,8]. Hyperactivated motility is also relevant and necessary for capacitation and the acrosome reaction, the preliminary stage of fertilization [8]. A decrease in sperm motility considerably reduces the ability of spermatozoa to reach the site of fertilization and to penetrate the oocyte, consequently decreasing male fertility.

In recent years, many researchers have focused on possible factors leading to male infertility, particularly leading to asthenozoospermia, revealing the existence of many cellular and molecular players, with more than 4000 genes thought to be involved in sperm production and as regulators of different aspects of sperm development, maturation, and function [9,10,11,12,13,14,15,16,17,18,19] (Figure 1). All those genes can potentially cause male infertility if mutated. Disfunction in some of those genes has already been identified in infertile men patients (Table 1). In addition to the genes that are known to play a role in the pathophysiology of male infertility, noncoding RNAs have been increasingly associated with male infertility. Noncoding RNAs are RNA molecules that are not translated into proteins but have multiple functions in mammalian spermatogenesis, including the development of the testes, differentiation of spermatogonial stem cells, and regulation of spermatocyte meiosis [20,21,22,23,24,25,26,27].

In this review, we briefly go through the sperm flagellum’s typical morphology and compile some of the most relevant information regarding the genetic factors involved in male infertility with a focus on sperm immotility and on genes related to sperm flagellum development, structure, or function. 

## 2. Sperm Flagellum

The sperm flagellum is responsible for sperm motility and contains both the site of energy production and the propulsive apparatus for the cell. The sperm flagellum is divided into four major regions, the connecting piece (CP), the midpiece (MP), the principal piece (PP), and the endpiece (EP). The CP is mainly composed of the segmented columns (SC), the proximal centriole (PC), distal centriole (DC), and the basal plate. The MP includes the axoneme (Ax; which extends for all flagellum), which is surrounded by the outer dense fibers (ODFs) and mitochondrial sheath (M). The MP and PP are separated by the annulus (An). The proximal PP contains the Ax surrounded by the ODF and the rings of the fibrous sheath (FS). In the distal PP, the Ax is encircled by the FS, and the EP only contains the Ax (Figure 2).

The sperm flagellum Is formed during a complex process named spermatogenesis that takes place within the seminiferous tubule and lasts approximately 64 days [28]. Spermatogenesis involves three main phases: (1) a proliferative phase where spermatogonia divide to replace their number (self-renewal); (2) a meiotic phase where spermatogonia differentiate into cells that lose contact with the basement membrane and commence the process of meiosis, which finishes with the formation of round spermatids (haploid cells); and (3) a spermiogenesis phase in which spermatids undergo a profound metamorphosis to become mature spermatozoa. The last step in spermatogenesis is often divided into two substages: spermiogenesis and spermiation. Spermiogenesis starts right after spermatocytes complete the second meiotic division, during which spermatids undergo a complex cytodifferentiation until the formation of elongated spermatids that, when fully maturated, are released from the seminiferous epithelium. The process of releasing the elongated spermatids is known as spermiation [29].

Intraflagellar transport (IFT) is a bidirectional process via which molecules are constantly transported, such as proteins needed for flagellum formation and selected signaling molecules, also enabling the recycling of turnover products. All molecules transported by the IFT process are collectively known as IFT particles (further reviewed by Taschner and Lorentzen [30], Gonçalves and Pelletier [31], Reiter et al. [32], Lechtreck [33], and Rosenbaum and Witman [34]). However, Agustin and coworkers found that, although IFT is critical for sperm development, it is not present in mature sperm and, thus, not needed for sperm maintenance [35]. 

There is another transport mechanism during flagellar extension at spermiogenesis, i.e., intramanchette transport (IMT). Structurally, the manchette is a transient skirt-like bundle of microtubules that is only present during spermatid elongation, being responsible for acrosomal vesicle and nuclear elongation. This transient structure consists of a dense perinuclear ring and inserted microtubules. The manchette is strategically located at the nuclear–cytoplasmic interface, which facilitates the Ran–GTPase-mediated nucleocytoplasmic trafficking of somatic histones [36,37]. The manchette contains important proteins, such as the Hook1 protein, which is necessary for the correct positioning of microtubular structures within the haploid germ cell. These proteins are involved in both intramanchette transport and nucleocytoplasmic exchanges. IMT is as temporary manchette itself, being only present during the spermatid elongation steps. In addition to having a role in the nucleus and acrosomal vesicle remodeling, the IMT cooperates with IFT in the development of the sperm CP and tail [38,39,40]. The *AXDND1* gene was found to be localized in the manchette, and, when mutated, the late spermatids showed head deformation, outer doublet microtubule deficiency in the axoneme, and loss of corresponding accessory structures, including ODFs and mitochondria sheath. Furthermore, *Axdnd1* knockout mice males are sterile with a reduced testis size caused by increased germ cell apoptosis and sloughing, exhibiting phenotypes consistent with oligoasthenoteratozoospermia [41].

### 2.1. Axoneme

The axoneme extends throughout the flagellum, being the structural core of the flagellum and the major component responsible for sperm motility. Structurally, the axoneme is highly conserved, and its structure is the same in all motile cilia/flagella [42]. The axoneme is a hollow cylinder microtubule-based structure composed of two central microtubules surrounded by nine peripheral microtubule doublets, forming the 9d + 2s microtubule pattern (nine doublets and two central microtubule pairs) (Figure 3). Each microtubule doublet (thereafter called a doublet) consists of an internal complete microtubule A, composed of 13 protofilaments, onto which a second external and incomplete microtubule B is attached, composed of 10 protofilaments [43,44]. Protofilaments are formed by α/β-tubulin heterodimers with a distinct structural polarity, where α-tubulins are exposed at the minus end and β-tubulins are exposed at the plus end. The regular and parallel orientation of tubulin subunits gives microtubules structural and dynamic polarity, with plus ends localized at the tip of the cilium, growing and shrinking more rapidly [45]. 

Microtubule A contains a pair of projections named dynein arms (DAs). The DA interacts with microtubule B of the adjacent doublet in an ATP-dependent manner, allowing doublet sliding and, thus, motility [46]. The DAs are designated by their position as inner (IDA) and outer (ODA) dynein arms. Each doublet is connected to the neighboring doublet through a nexin bridge, known as the dynein regulatory complex (DRC) [47]. The axoneme from motile cilia contains two central microtubules (C1 and C2), which are linked by a series of regularly spaced linkages (the central bridge) and surrounded by a fibrillar central sheath. This central structure constitutes the central pair complex (CPC), previously known as the central apparatus. Doublets are connected to the CPC through radial projections called radial spokes (RSs).

Dyneins are a large family of motor proteins known as ATPases associated with diverse activities (AAA+) [48,49]. They are responsible for converting the energy gained from ATP hydrolysis into mechanical work, which enables the microtubule sliding and, thus, the flagellar movement [50,51]. 

The DAs are regularly placed along microtubule doublets. The ODAs are organized at intervals of 24 nm and believed to be responsible to regulate the ciliary beat frequency (CBF). The ODAs are controlled by external factors such as phosphorylation, variations in Ca^2+^ concentrations, and alterations in the redox state [52]. At least 13 genes are related to ODA, such as DNAH5, DNAH9, and DNAH6 [49,53]. In contrast, the IDAs are docked to doublets precisely in a 96 nm axonemal repeat and were shown to control the ciliary/flagellar bending [54]. IDAs are composed of proteins derived from at least 15 genes, including DNAH3, DNAH7, and DNAH11 [49,53]. 

Dynein motors are synthesized and preassembled in the cytoplasm and then transported into the cilia by the IFT machinery (further reviewed by Desai et al. [55]), being anchored onto doublets by proteins that form high affinity docking sites, such as CCDC103 [56], CCDC114 [57], ARMC4 [58], and TTC25 [59]. Because dyneins are large polypeptides, their synthesis takes time (about 13 min per dynein molecule/per ribosome) and requires the support of various protein factors termed dynein axonemal assembly factors (DNAAFs) [49,55]. DNAAFs are known to play a role in substrate recognition for folding and stabilization of the catalytic dynein subunits [55]. The list is still growing, but at least eight genes were proposed to originate proteins that act as a chaperone and in the cytoplasmic assembly of axonemal dyneins, such as *DNAAF1* (previously known as *LRRC50*), *HEATR2*, and *DNAAF3* [60,61,62]. 

The DRC Is involved in interdoublet sliding and in mediating signals between different axoneme components. The DRC, thus, acts as an intermediate in the communication between the CPC–RS complex and DA [47]. The DRC has two distinct parts, the base plate, via which the DRC is attached to doublet microtubule A, and the linker, which allows the connection between doublet microtubule A and the microtubule B of the following doublet [47]. At least 11 DRC proteins have been identified [63,64,65,66]. Studies using Chlamydomonas mutants lacking DRC showed that, in addition to regulatory functions, DRC plays a critical role in mediating structural interactions among DA, doublet microtubule A, and the RS [67,68,69]. Furthermore, DRC proteins 3, 4, and 7 were suggested to stabilize the binding of IDA isoforms, whereas components 1 and 2 may be part of the binding site for both RSs and certain IDA subspecies [70]. 

Radial spokes present a T-shaped structure, composed of a stalk that is anchored on doublet microtubule A, and a head, which is assumed to have temporary contact with the CPC central sheath [71,72]. In Chlamydomonas, at least 23 distinct polypeptides are known to compose RS. These are named RS proteins (RSP) 1–23 that present a combined molecular mass of approximately 1200 kDa [73]. However, several of them do not show homology to proteins of known structure or function, and knowledge is still scarce regarding these proteins. Nevertheless, some key functions have already been attributed. For instance, RSP3 is known, together with the LC8 protein (dynein light chain), to be critical in RS stalk constitution, and to assist the dock of the stalk to its RS docking complexes [74,75]. Furthermore, it is known that RSPH3 is a protein kinase A-anchoring protein (AKAP) [74,76] and, as other AKAP proteins, it can bind to protein kinase A (PKA) substrates. The connection between AKAP proteins to PKA substrates enables the activation of the PKA catalytic subunit and, consequently, leads to the rise in cyclic adenosine monophosphate (cAMP) levels [77,78,79]. The presence of AKAP proteins in RS composition may suggest the existence of phosphorylation/dephosphorylation mechanisms, likely for DA regulation [80]. Studies in Chlamydomonas have shown that RS are partially assembled in the cytoplasm. The mechanism of assembly of all RS proteins is not known. Regarding the 12S RS complex, which includes RS proteins 1–7 and 9–12, it is known that they assemble in the cytoplasm. The 12S RS complex is then transported, by the IFT, to the tip of the flagellum/cilium, where it is combined with other RSPs to form the 20S mature RS complex (that contains all RS proteins) on the axonemal microtubules [81,82].

The RS head and the CPC are spaced apart In the straight axoneme, but the two structures come into contact when the axoneme bends, and it was proposed that the distance between the CPC and RS head is critical for proper movement [72]. Moreover, the RS head architecture and biochemical composition seem to have vital importance in the mechano-signaling process with the CPC, with mutations that destabilize the RS head resulting in impaired cilia motility [72].

About CPC microtubules, studies in Chlamydomonas have shown that 24 proteins compose CPC microtubules, with 10 being unique to microtubule C1 and seven being unique to microtubule C2, suggesting that CPC microtubules are biochemically and structurally distinct [83]. Although the structure of the axoneme is thought to be well conserved throughout species, specific knowledge regarding the mammalian CPC proteins is still limited, and only a few mammalian CPC proteins are known, including SPEF2, HYDIN, CFAP221, CFAP54, SPAG6, SPAG16, SPAG17, and PCDP1 [84,85]. Concerning CPC assembly, CPC microtubules are not anchored or formed from basal body microtubules. How CP microtubules assemble is not clear, but studies have suggested that CPC microtubules self-assemble without requiring a template within the cilium/flagellum, and that the tubulin used for CPC assembly is imported directly from the cytoplasm [86]. A study from Nakazawa and coworkers corroborated the previous findings that there is no organizing center for the CPC assembly and showed that the space size within the axoneme is an important factor that directs CPC formation in the axoneme. The authors suggested that, when the CPC precursors are present in the axoneme, the CPC may form spontaneously without interacting with any template or RSs. In such a case, only the available space and the amount of precursors may limit the assembly of CPCs [87].

The putative serine–threonine kinase fused (FU) protein and the kinesin protein KIF27 were proposed to be involved in the recruitment or activation of CPC proteins, such as SPAG16 and PCDP1, thus being required for CPC assembly in motile cilia [88]. However, the complete cascade of events via which Fu/Kif27 leads to CPC assembly is unknown.

Anomalies in any of the components that are needed for the formation or function of the axoneme impair sperm motility and, consequently, lead to infertility [8,42,89,90,91].

### 2.2. Connecting Piece

The CP (also known as neck piece) is a complex structure, with about 0.5 µm in length that connects the sperm head to the sperm flagellum. It is composed of the nuclear base, basal plate, capitellum (or capitulum), segmented (or striated) columns, and centrioles [92]. 

Round spermatids contain the typical centrosome complex with two orthogonally arranged centrioles, which are critical to guide sperm development. The distal centriole (DC) serves as a template for assembling the axoneme (a microtubule-based cytoskeletal structure that forms the core of the flagellum), and the other, called the proximal centriole (PC) [93], appears to contribute to the nuclear shaping [94]. After the formation of the flagellum in the early spermatid, the PC increases the length by the addition of material to its free distal end, and the centriolar elongation continues throughout the differentiation of the CP. Ultrastructural studies have observed that the fine structure of the material added to the distal end of the centriole in elongation is distinct from that of the original centriole. Hence, this structure was named the centriolar adjunct to denote that it is not simply a prolongation of the juxtanuclear centriole but is a new structure of slightly different internal organization [95]. The centriolar adjunct is a dynamic structure, which decreases during spermatozoa maturation and disappears partially or completely in the mature spermatozoa of different organisms. In humans, it was shown that the disassembly of the centriolar adjunct is required for the functional maturity of human spermatozoa, and an incomplete disassembly of the centriolar adjunct is associated with infertility [96].

The apical region of the CP contains the basal nuclear region that presents a redundant nuclear envelope with numerous nuclear pores, behaving as a calcium store. The nuclear lamina of the basal nuclear region thickens to form the basal plate. This is connected to a dense fibrillar material, named the capitellum, composed of rigid proteins, from which the striated columns are formed. The capitellum is situated right below the basal plate [95]. The striated columns encircle the PC, firmly linked to the capitellum [93,97]. Below the PC, there is a pale region containing a modified DC. During spermiogenesis, the DC gives rise to the axoneme, and then loses the triplet structure and acquires the form of a cone, with the apex distally located. The axoneme is present from the distal region of the atypical DC. Until recently, it was widely accepted that the DC was disintegrated during late spermiogenesis after axoneme formation, leaving only a few remnants in mature spermatozoa. However, recent studies reported the presence of a functional, yet highly modified and atypical DC [98]. It is supposed that, after fertilization, both centrioles, the typical PC and the atypical DC, are involved in aster formation. Below the DC, a microtubule organizer center polymerizes the central microtubule pair of the axoneme (further reviewed by Avidor-Reiss, Mazur, Fishman, and Sindhwani [94]).

At the basolateral nuclear region of the sperm head, the cytoplasmic membrane is cramped by a posterior ring, separating the head compartment from the flagellum compartment. Below the nuclear ring, the dense basal nuclear lamina creates a basal nuclear fossa, where the capitellum attaches. The sperm nuclear envelope at the head region does not contain nuclear pores, as sperm chromatin is firmly packaged and inactive, and the absence of nuclear pores is believed to protect against hydrolytic enzymes released during the acrosome reaction. The exception to this is the basal nuclear region at the CP [99].

The capitellum gives rise to distal striated columns [99]. The capitellum is a curved plate-like disc localized at the beginning of the CP to which the sperm head is firmly associated by its association with the basal plate. In the mature spermatozoon, at the distal end of the striated columns, each column is connected with one of the nine outer dense fibers (ODFs). The striated columns are formed separately, and then become linked at a late stage in spermiogenesis, while the ODF only attaches to the distal ends of the striated columns in the mature spermatozoon [100]. Although linked to striated columns, the ODFs are not present at the CP. Cryo-electron tomography has shown that, within the striated columns, there are areas with lower (pale bands) and higher (bark bands) density, which previously were thought to be a staining artefact. Within the pale bands, there are molecular linkers that connect to the adjacent dark bands, which likely have some role in regulating the bending of the flagella, instead of being just structural elements [100]. However, how these linkers participate in the regulation of bending is still not known. Furthermore, Yuan and coworkers have shown that striated columns are required for normal development of the CP, specifically for the proper attachment and alignment of the sperm head and MP. They demonstrated that the SPATA6 protein was exclusively expressed in the CP of spermatozoa. By inactivation of the mice *Spata6* gene, these authors observed that mice were sterile due to the presence of acephalic spermatozoa, caused by a complete lack or partial development of the CP. As the major ultrastructural defect found in these null Spata6 mice was the complete lack of the capitellum and/or striated columns, the authors proposed that Spata6 is required for the formation of the capitellum and striated columns and, consequently, for the normal assembly of the sperm flagellum [101]. Moreover, recent work suggested that the SPATA20 protein is essential for sperm head–tail conjunction formation in humans and that it colocalizes with SPATA6. The presence of a null pathogenic variant in *SPATA20* gene impairs the expression of both *SPATA20* and *SPATA6*, leading to male infertility related to acephalic spermatozoa syndrome [102]. 

### 2.3. Midpiece 

The MP is about 3.5–5 µm in length and is localized after the CP. The MP includes the axoneme, which extends throughout the flagellum. In the MP, the axoneme is surrounded by nine ODFs and, more externally, by a mitochondrial sheath in the form of a helix. The human sperm mitochondria are positioned in four helices, and each helix is composed of 11–15 gyres with, on average, two mitochondria in each gyre [92].

The ODF begins at the distal region of the striated columns. The end of the mitochondrial sheath is marked by the presence of a proteinaceous ring, the annulus. The ODFs are formed during spermiogenesis and localize to the PP and PPP of the sperm tail, on the outside of the axoneme [103]. The ODFs have an important role in the flagellum. ODFs provide elasticity [104] and enhance the tensile strength of the flagellum that is required to overcome the shear forces found in the female reproductive tract [105], and they also protect the integrity of the axoneme to promote sperm motility [106]. Ultrastructural defects in ODF were reported in the spermatozoa of asthenoteratozoospermic men (abnormal sperm morphology and reduced motility) [91,106,107]. 

Mammalian ODFs consist of at least 14 polypeptides, such as ODF1-4, SPAG2, SPAG4, and SPAG5, which are important for proper sperm function [108,109,110,111,112,113,114,115,116,117,118,119]. The role of all of those proteins in sperm function is not clear. However, the expression levels of ODF1–4 were frequently downregulated in asthenozoospermic samples [106]. Moreover, it is known that ODF1 is involved in the correct arrangement of the mitochondrial sheath and ODF, and it was shown to be essential for the rigid junction of the sperm head and tail [120,121]. Moreover, the ODF2 protein was proposed as critical for the formation and stabilization of ODF structures [106,112,122]. 

During spermatogenesis, sperm mitochondria suffer several changes. Mitochondria in spermatogonia and early spermatocytes present the typical morphology observed in most somatic cell types. As spermatogenesis progresses and the cytoplasm reduces, the mitochondrial matrix becomes denser and mitochondria decrease in number, with a concomitant increase in mitochondrial activity [123,124]. In the mature human spermatozoon, mitochondria are positioned in four helices. Each helix is composed of 11–15 gyres with an average of two mitochondria per gyre. Sperm mitochondria are more resistant than somatic mitochondria due to the presence of a capsule. The sperm mitochondrial capsule is formed by disulfide bonds between cysteine- and proline-rich proteins, such as the sperm mitochondria-associated cysteine-rich protein (SMCP), phospholipid hydroperoxide glutathione peroxidase (PHGPx), and mitochondrial selenoprotein [125]. 

The sperm mitochondrial capsule makes mitochondria resistant to the hypoosmotic environment that the sperm encounters during the male-to-female reproductive tract transition, and to the stretching and compression imposed by the flagellar beat [92,126]. Furthermore, evolution also gave sperm mitochondria other particularities that distinguish them from somatic mitochondria, adapting them for sperm motility. In this sense, sperm mitochondria can use lactate as an oxidative substrate, and they contain specific isoforms of proteins and isoenzymes, such as cytochrome-c, making them more efficient and adaptable to different conditions [127]. 

The localization of mitochondria at the beginning of the axoneme, the most energy-demanding structure of the flagellum, means that a major role of mitochondria is to provide the energy required for proper sperm motility. However, it is an ongoing debate if mitochondrial oxidative phosphorylation is indeed the major source of energy for sperm motility, or if the main energy source for sperm motility comes from glycolysis driven by glycolytic enzymes that are linked to the FS (reviewed by Ford [128], du Plessis et al. [129], and Losano et al. [130]). Additionally, mitochondria play other important roles in sperm, including the generation of metabolites in the tricarboxylic acid (TCA) cycle. These metabolites have diverse downstream applications, including in signaling, in epigenetic regulation, and as intermediates in the synthesis of pyrimidine, purine, amino acids, and fatty acids [131]. Furthermore, mitochondria are the reactive oxygen species (ROS) generator in the spermatozoon, and, at physiological concentrations, ROS are the trigger for several important mechanisms, such as sperm capacitation, hyperactivation, the acrosome reaction, and oocyte fusion [132,133,134,135]. Additionally, a recent study suggested that mitochondria may also participate in acrosomal vesicle formation, as Golgi proacrosomal vesicles acquire mitochondrial proteins [136].

Mitochondrial genetics is highly complex and still not fully understood. Nevertheless, numerous mitochondrial genes have been related to male infertility (reviewed by Demain et al. [137] and Vertika et al. [138]). For instance, a 4977 bp deletion in the mitochondrial genome was found to be correlated with spermatogenic failure [139], and genetic variations in two mitochondrial genes (*MT-ND4* and *MT-TL1*) were proposed as a cause of asthenozoospermia in China [140]. SDHA is a component of the succinate dehydrogenase (SDH) complex and plays a critical role in the mitochondria. Recently, it was reported that the *C. elegans* SDHA ortholog *SDHA-2* is essential for male fertility; *SDHA-2* mutants produce dramatically fewer offspring due to defective sperm activation and motility, and they have aberrant mitochondrial morphology and dynamics, as well as a disrupted redox balance [141]. Thus, *SDHA-2* is a newer candidate hypothesized to cause male infertility. 

In the MP, a cytoplasmic droplet is sometimes observed that consists of residual cytoplasm not fully removed during spermiation due to incomplete extrusion from Sertoli cell cytoplasmic pockets [142,143]. Residual cytoplasm is considered an anomaly only when in excess, i.e., when it exceeds one-third of the sperm head size [99]. Human spermatozoa that present a cytoplasmic droplet around the MP, in contrast with excess residual cytoplasm, are still considered normal, and a small cytoplasmic droplet does not interfere with motility or fertility. Even without a cytoplasmic droplet, the apical region of the MP contains a small region with a few small vesicles. This region is thought to be the major site of water influx and cell volume regulation, and the small vesicles are important when spermatozoa face hypoosmotic challenges [143,144]. 

### 2.4. Principal Piece

The major part of the human sperm flagellum is the PP with about 44–50 µm in length [145]. The PP is separated from the MP by a septin-based ring structure, located at the end of the mitochondrial sheath, named the annulus or Jensen’s ring. The annulus is firmly adherent to the flagellar membrane at the beginning of the PP and is not easily observed [104]. The annulus is formed before the axoneme starts to extend from sperm, during late spermiogenesis, and before mitochondrial sheath formation. During sperm tail elongation and differentiation, it migrates along the axoneme toward its final position at the junction of the MP and the PP [146]. Thus, the annulus acts as a morphological organizer guiding the growth of the flagellum and the alignment of the mitochondria along the axoneme [146]. Moreover, it has a direct role in restricting the diffusion of membrane proteins, thus acting as a sperm diffusion barrier [147]. The mammalian sperm annulus is composed mainly of proteins from the septin family, namely, septins (SEPT) 1, 2, 4, 7, and 12 [148,149,150,151,152,153,154,155]. The absence of annulus and the disfunction in septin proteins, particularly in SEPT4 and SEPT12, is associated with an abnormal sperm flagellum and immotility [152,156,157,158,159,160,161].

The PP is distinguished from the other parts of the flagellum by the presence of the FS, which surrounds the ODF and the axoneme. The FS consists of two longitudinal columns, at the plane of the central microtubules, connected by a series of semi-circumferential ribs, which form a ring around the axoneme [92]. The FS is attached to ODFs 3 and 8 in the proximal part of the PP. At the distal part of the PP, the FS becomes thicker causing ODF termination, with the FS becoming associated with doublets 3 and 8. FS assembly progresses from a distal to a proximal direction along the axoneme throughout spermiogenesis [162]. To date, more than 20 FS-related proteins have been documented [163]. Among those, AKAP proteins are known as the main structural components of the FS, specifically AKAP3 and AKAP4 [162]. FS proteins are widely distributed across the cytoplasm of spermatids; however, when the sperm flagellum is formed, they specifically localize at the FS [164,165,166]. It was proposed that FS proteins are synthesized in the cytoplasm and remain in an immature form. Then, they are transported by the IFT to the FS assemble site, where they are converted into a mature protein form by the action of proteolytic enzymes [164,167]. A recent study in a mouse model demonstrated that IFT74, an IFT protein, is essential for sperm flagellum formation, suggesting that it has an important role in the assembly of the core axoneme. This was shown by the fact that mutant sperm presented very short or absent tails, with spermatids also presenting a variety of axonemal abnormalities, such as complete absence of the axoneme, disorganized microtubules, and abnormally formed axonemes without the central microtubules. Moreover, in the mutant mice, the testicular expression of the immature form of the AKAP protein (proAKAP4) was observed to be the predominant form, which suggests that, due to disruption of IFT, the pro-AKAP4 is not transported to the FS assemble site; thus, the AKAP4 protein is not processed into its mature form [167]. 

Some important roles were attributed to the FS. Specifically, it was assumed that the FS influences and modulates flagellar bending and flagellar beating, likely by being involved in the cAMP signaling pathway. The AKAP4 protein was shown to act as an important regulator of the cAMP/PKA and PKC/ERK1/2 signal pathways, which are both important to regulate processes that lead to fertilization, namely, the forward and hyperactivated motility and the acrosome reaction [8,168,169]. In addition, FS anchors glycolytic enzymes that provide energy for sperm motility [162,170]. Anomalies in the FS are associated with the absence of sperm motility and, thus, with infertility [89,91,171,172,173,174,175,176]. Dysplasia of FS (DFS) is a well-defined syndrome characterized by marked hypertrophy and hyperplasia of the FS, often associated with annulus absence; consequently, an abnormal FS invades the MP. Furthermore, the absence of CPC and DA have also been observed [8,91]. Although it was proposed that DFS might have a genetic base, no consensus exists regarding it. Indeed, no association between DFS and defects in genes that code for the AKAP3 and AKAP4 proteins were observed in patients with DSF [89].

### 2.5. Endpiece

In human spermatozoa, the endpiece starts when the FS completely disappears. Thus, the EP only contains axonemal components. At the proximal part of the EP, the axoneme structure is kept. Similarly, to the cilium, at more distal regions, the axoneme structure becomes disorganized, and only single microtubules are present (singlet region) [177]. Studies in human sperm using cryo-electron microscopy found that the microtubule number in the singlet region was greatly variable, ranging from two to 14 single microtubules, with no doublet microtubules observed closer than 1.6 µm from the tip. Furthermore, the authors showed that, in the endpiece of sperm, doublet microtubules can split into two singlet microtubules [178,179].

At the EP, a novel protein complex was identified and named tail axoneme intra-lumenal spiral (TAILS). TAILS binds directly to the 11 protofilaments on the internal microtubule wall, assembling into an array along protofilaments. The authors hypothesized that TAILS has a role in stabilizing microtubules and/or enabling rapid swimming, and it plays a role in controlling the swimming direction of spermatozoa [179].

No specific role in sperm motility was assigned to the endpiece. However, recently, a study proposed a role in sperm propulsion by showing that spermatozoa with a short, mechanically inactive region at the flagellum end swim faster and more efficiently than those without [180]. Neal and coworkers proposed that the optimal inactive fraction should be 5% of the human flagellum tip. This finding could have important consequences in human-assisted reproduction, i.e., in the development of a new methodology for improving the analysis of flagellar imaging data [180].

## 3. Motility Mechanisms 

Flagellar beating consists of an undulating wave propagating from the base to the tip of the flagellum [181]. As stated above, the main component responsible for the ciliary and flagellar motility is the axoneme (with the 9d + 2s structure), known as the “motor” complex. In particular, ODAs are responsible for the control of the beat frequency and provide much of the power required for movement [52]. On the other hand, IDAs are responsible for the control of the size and shape of the bending [54]. When activated, DAs and the associated doublet microtubule A move in the proximal direction relative to doublet microtubule B of the next adjacent doublet. For proper bending, DAs should follow a switching mechanism that controls the activity of the dynein motors on different doublets at opposite sides of the bending axis. This means that some DAs on one side of the axoneme are responsible for generating the forward bend, whereas other DAs produce the reverse bend [182]. Recent cryo-electron tomography data showed that contacts between neighboring dyneins change during the power stroke cycle of dyneins, suggesting the existence of an elegant and complex mechanism for coordinated dynein activity, where an ATP-dependent conformational change in one dynein is propagated to the neighboring dynein [183].

The CPC and RS were proposed to be essential for the regulation of DA activity and, thus, doublet microtubule sliding [71,72]. This proceeds likely through interactions among specific CPC and RS kinases and phosphatases, which are believed to be involved in flagellar motility by participating in the cAMP pathway [73,184,185]. It was suggested that mechanical interactions between the CPC and RS are converted into biochemical signaling that may alter the phosphorylation state of the DA, which then regulates doublet microtubule sliding [70,186]. Alternatively, the CPC and RS may be involved in converting symmetric bends into the asymmetric waveforms required for forward swimming, and in the release of ATP inhibitors in a controlled manner [83,186,187]. Moreover, the CPC may function as a distributor to provide a local signal to the RS that selectively activates subsets of DA [184,188]. However, these are speculations, and the exact mechanism of how this regulation occurs is not known; it is also unknown if the CPC and RS may have an individual or interactive role in ciliary/flagellar motility. Nevertheless, it is well established that the axoneme is the major player in ciliary/flagellar motility, and that it requires coordinated DA activation and inactivation, which is highly regulated by the CPC, RS, and DRC.

It was proposed that age may affect ciliary motility, since, in airway epithelium, the CBF was observed to differ with age, with the CBF in children being significantly greater than in adults [189,190]. However, data are not consensual for sperm flagellum motility. Some authors referring to have not observed differences in sperm motility with aging [191], while others observed a reduction with ageing [192]. Nevertheless, these studies are complex, as lifestyle habits and environmental conditions may affect the behavior of the cells and, thus, affect results. 

Regarding human sperm motility, recent 3D studies have shown that the sperm flagellum beats anisotropically and asymmetrically. Anisotropically means that the wave characteristics in each transversal direction (i.e., two beating planes, a plane perpendicular to the FS and a plane parallel to the FS) differ significantly. Asymmetrically means that the beating planes do not have a left–right symmetry (i.e., the bending is larger for one side and smaller for the other side) [193]. 

The origin of those beating characteristics is not fully understood but may rely on the flagellum structure. The sperm flagellum contains accessory structures, namely, the ODF and FS, which were proposed to be the reason for the mechanical anisotropy along the flagellum. The major bending was observed where the beating plane is perpendicular to the FS. In contrast, where the beating plane lies parallel to the FS, bending may be obstructed and is, thus, smaller [193]. Furthermore, the anisotropic bilateral localization of the voltage-gated proton channel Hv1 and the symmetrical, quadrilateral localization of the sperm cation channel CatSper, both involved in Ca^2+^ entry and regulation, were also proposed to be involved in the control of sperm rotation, particularly in the anisotropic beat generation [193,194].

The control and regulation of motility is a complex process in both cilia and flagella. In spermatozoa, the capability to move is acquired during the transit through the epididymis, where they mature. However, spermatozoa only become motile after ejaculation. 

Several molecular mechanisms are known to be involved in sperm motility [8]. A known molecular pathway involved in the acquisition of motility capacity in the epididymis is the relationship among protein phosphatase methylesterase 1 (PPME1), post-translational modifications of protein phosphatase 2 catalytic subunit α (PPP2CA), and the serine/threonine protein kinase (GSK3). Conjointly, they trigger a complex series of molecular reactions that culminate in the inactivation of serine/threonine protein phosphatase 1 (PP1), which allows sperm to become motile (reviewed by Freitas et al. [195]). 

An example of those complex interactions was given by the study of Koch and coworkers, which showed that, in the last portion of the epididymis, GSK3 is inhibited by Wnt signals, allowing the inactivation of PP1. PP1 is kept in an active state by phosphorylation of its inhibitory subunit PPP1R2 through the action of GSK3 [196]. On the contrary, when Wnt signals block GSK3, the PPP1R2 is not phosphorylated, and PP1 remains inactive, which was observed to be critical to enabling sperm to acquire motility capacities without becoming motile in the epididymis [197]. 

In the female reproductive tract, spermatozoa are exposed to a complex environment, where molecular signals promote different mechanisms that regulate sperm motility crucial to uterine migration from the cervix to the uterus and fertilization. To travel through the female tract, spermatozoa require the assistance of the seminal plasma, which is produced by different male sex organs (including the testis, the epididymis, the prostate, the Cowper (bulbourethral) and Littre’s (periurethral) glands, and the ampulla of the ductus deferens). Seminal plasma is a very rich fluid that contains a plethora of sugars, oligosaccharides, glycans, lipids, proteins, inorganic ions (calcium, magnesium, potassium, sodium, and zinc) and small metabolites. Changes in the seminal plasma composition have already been associated with infertility [10,198,199,200,201,202,203,204,205]. Moreover, seminal exosomes have been identified in human seminal plasma. Exosomes are membranous nanovesicles of endocytic origin that carry and transfer regulatory bioactive molecules and mediate intercellular communication between cells and tissues [206]. Research on seminal exosomes has increased throughout the years and several pieces of evidence have been given that associated seminal exosomes with infertility [207,208]. 

An example of the relevance of seminal plasma is given by the Semenogelin (SEMG1) protein. As stated above, spermatozoa only become motile after ejaculation. SEMG1 is a seminal plasma protein that inhibits sperm progressive motility; immediately after ejaculation, it is hydrolyzed by the prostate-specific antigen (PSA), thus removing SEMG1 from the sperm surface. This allows sperm to become motile in the vaginal canal and capacitation to proceed during uterine migration [7,209]. 

At the beginning of the sperm journey throughout the female tract, the first master regulator is the cervical mucus composition, i.e., the pH, which can impair spermatozoa motility if too acid [210]. In addition, the mucus molecular composition might also affect the sperm journey. So far, more than 800 proteins have been identified to compose the cervical mucus [211]. How these proteins interact with seminal plasm proteins remains unknown. However, a recent study suggested that genomic compatibility at the cervix might also be involved in couple fertility, which may open the doors to a new research field in reproductive biology. The authors genotyped all seven classical HLA genes of participants, diluted cervical mucus samples from nine women, and combined them with washed spermatozoa aliquots from eight men. All possible male–female combinations were tested for spermatozoa performance. They observed that spermatozoa motility was different depending on male–woman HLA compatibility, with combinations that had low HLA divergence having higher sperm motility. This suggests that the genotype of the cervical mucus may affect the couple’s compatibility, and that the cervical mucus could mediate the sexual selection of spermatozoa from immunologically compatible males [212].

The regulation of ion balance is also essential for sperm motility, particularly Ca^2+^, H^+^, HCO_3_^−^, Na^+^, and K^+^ [213]. Calcium is vital to induce hyperactivated motility during sperm capacitation in the uterine milieu. In mammals, CatSper is the sperm-specific Ca^2+^ channel complex responsible for flagellar Ca^2+^ entry [214]. Although other Ca^2+^ channels exist in the sperm flagellum, CatSper is the predominant Ca^2+^ channel and the major modulator of the physiological processes of sperm hyperactivation, sperm capacitation, chemotaxis toward the oocyte, and the acrosome reaction [214]. Mutations in Catsper channels have been associated with sperm immotility [215,216,217,218].

Recently, it was shown that CatSper channel complexes are arranged in zigzag rows along the longitudinal axis of wildtype human and mouse sperm flagella. The cytoplasmic EFCAB9–CatSperζ complex was proposed to be critical for the arrangement of the CatSper channel complex and the linear alignment within the longitudinal nanodomains; disruption of this complex results in fragmentation and misalignment of the zigzag rows and disruption of flagellar movement [219]. Furthermore, the C2CD6 protein contains a calcium-dependent, membrane-targeting C2 domain. This C2CD6 protein was observed to be associated with the CatSper calcium-selective, core-forming subunits. C2CD6-deficient sperm have a reduced number of the CatSper nanodomains in the flagellum, which results in male sterility due to defective hyperactivation and a failure to fertilize oocytes both in vitro and in vivo [220]. 

Sodium–hydrogen exchangers (NHEs), encoded by the *SLC9* gene family, import Na^+^ and export H^+^ from cells. These Na/K exchangers regulate the intracellular pH in different cell types, particularly in spermatozoa. Channel sNHE (or NHE10), encoded by gene *SLC9C1,* and channel NHA1 (or NHE11), encoded by the *SLC9B1* gene, are expressed specifically in the sperm flagellum [221,222]. Other ions, such as Cl^−^ and HCO_3_^−^ are also known to be indispensable for the capacitation process. This is accomplished through a cascade of reactions that involve the activation of soluble adenylyl cyclase (sAC), which leads to an increase in the intracellular levels of cAMP, by the conversion of ATP into cAMP. This event stimulates Ca^2+^ intake and the activation of PKA. Protein kinase A then triggers several phosphorylation cascades that lead to capacitation [223,224]. The cystic fibrosis transmembrane conductance regulator (CFTR) is a transporter of both ions, and its proper function is critical for sperm function, as specific inhibition of this channel has reduced cAMP production and membrane hyperpolarization [225]. It was shown that CFTR, together with TAT1, a member of the solute carrier family 26, number 3 (SLC26A3), work together in the regulation of the Cl^−^ and HCO_3_^−^ fluxes. These are required for activation of the sAC/PKA pathway during sperm capacitation [226]. In this regulation, and to become active in human spermatozoa, the CFTR channel requires PKA activity, with PKA being activated by a cascade of events dependent on HCO_3_^−^ ions. In this sense, it was proposed that the Na^+^/HCO_3_^−^ cotransporter (NBC) controls the initial intake of HCO_3_^−^ ions, followed by PKA activation. Thereafter, it is CFTR that controls the intake of HCO_3_^−^ ions [227].

The localization of ion channels is specific within sperm, corroborating the importance of the precise regulation of those channels for proper sperm function. For instance, the four CatSper Ca^2+^ channels [218], KSper (sperm-specific K^+^ channel, also known as SLO K^+^ channels) [228], and Hv1 [194] are found in the PP, and this location seems not to be random for different reasons. First, the asymmetrical organization of Hv1 is required for sperm rotation [193,194]. Second, all those channels are related to pH, thus suggesting a compartmentalized pH sperm motility regulation [213]. Third, those channels seem to interact with each other; specifically, SLO3 channels appear to play a key role in the activation of CatSper channels [229], and Hv1 and CatSper channels also interact to regulate human sperm physiology, likely via the regulation of pH, which then regulates Ca^2+^ fluxes [230]. In contrast, CFTR and TATI, master regulators of HCO_3_^−^ and Cl^−^, are colocalized at the sperm head and MP [226], likely because they are critical for the sAC/PKA signaling pathway, which is mitochondria-dependent [231]. Despite what is already known, the knowledge is still incomplete, and further studies are needed to better understand the interactions and the functions of the sperm ion channels. 

## 4. Conclusions and Future Challenges

So far, there is no effective therapy for sperm immotility; thus, to conceive a child, couples need to use assisted reproductive techniques (ARTs), particularly intracytoplasmic sperm injection (ICSI). Although some successful ICSIs performed with immotile spermatozoa have been reported, the fertilization rate after ICSI with immotile spermatozoa is usually low [232]. The success of the ICSI was proposed to be related to individual genotype [233] Moreover, sperm immotility is mostly caused by defects in multiple genes and/or by the interaction of multiple variants located in different genes; thus, using ART has the risk of transmitting the mutation to its offspring. Therefore, understanding which molecular players affect sperm motility, how they are correlated, how exactly proteins work, and how these variants may affect its function is essential to planning and developing effective therapies for infertility. 

Finding robust, reliable, and reproducible genomic biomarkers is essential to improve the diagnostics and management of male infertility. With the international announcement of the complete sequencing of the human genome, medical doctors and scientists believed that all genetic questions would be answered. However, despite the huge progress in genomics and amazing achievements, there are still several unanswered questions. Thus, finding biomarkers for male infertility is not always straightforward. It is important to keep in mind that the presence of gene mutation by itself may not represent a direct cause–effect relationship, especially for multifactorial diseases such as male infertility, which may be caused for several problems during the complex spermatogenesis process, as well as other pathologies and infections. The knowledge regarding the genetic etiology underlying male infertility is still far from complete. Neither the role of modifiers genes, nor how the transcriptome, proteome, and epigenomic influence the clinical phenotype is fully understood. 

Induced pluripotent stem cells (iPSC) and genome editing technology show great promise for biological and therapeutic applications for male infertility. Remarkable work conducted by Hayashi et al. reported that iPSC could be differentiated into fertile spermatozoa and oocytes via primordial germ cell-like cells [234]. However, the studies are still limited and controversial. Several barriers, including ethical issues, the risk of transmitting genetic insults to the offspring during in vitro culture of stem cells, and the safety and efficacy of genome editing, must be overcome. Hopefully, a platform for modeling male infertility using stem cells will be created, and improvements in the knowledge regarding the mechanisms related to male infertility will be achieved. 

## Figures and Tables

**Figure 1 genes-14-00383-f001:**
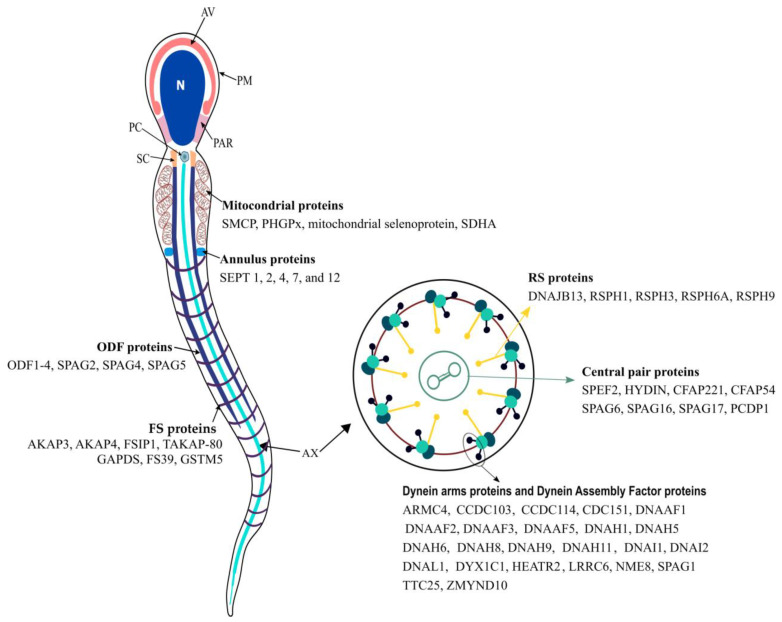
Schematic view of a human sperm cell and main proteins associated with the main structures from sperm flagellum. The sperm cell is morphologically divided into two parts: the head and the flagellum. The head is composed of the plasma membrane (PM; which surrounds all sperm cells), acrosomal vesicle (AV), nucleus (N), and postacrosomal region (PAR; which connects the head to the flagellum). The sperm flagellum is further divided into four major regions, the connecting piece, the midpiece, the principal piece, and the endpiece. The connecting piece is composed mainly of the segmented columns (SCs), the proximal centriole (PC) and distal centriole, and the basal plate. For simplicity, the basal plate and distal centriole are not represented. The midpiece includes the axoneme (Ax; which extends for all flagellum), which is surrounded by the outer dense fibers (ODFs) and mitochondrial sheath (M). The Ax is composed of nine axonemal doublets microtubules (also known as the peripheral doublets) linked by the dynein regulatory complex (DRC) and connected by radial spokes (RS) to a single pair of central microtubules, which are surrounded by a fibrillar central sheath, constituting the central pair complex (CPC). The peripheral doublets are composed of microtubule A and microtubule B. From each microtubule A arise two dynein arms: the outer (ODA) and inner (IDA) dynein arms. The sperm midpiece and principal piece are separated by the annulus (An). The proximal principal piece contains the Ax surrounded by the ODF and the rings of fibrous sheath (FS). In the distal principal piece, the Ax is encircled by the FS, and the endpiece only contains the Ax, which gradually becomes disorganized and loses its 9d + 2s conformation. See text for further details.

**Figure 2 genes-14-00383-f002:**
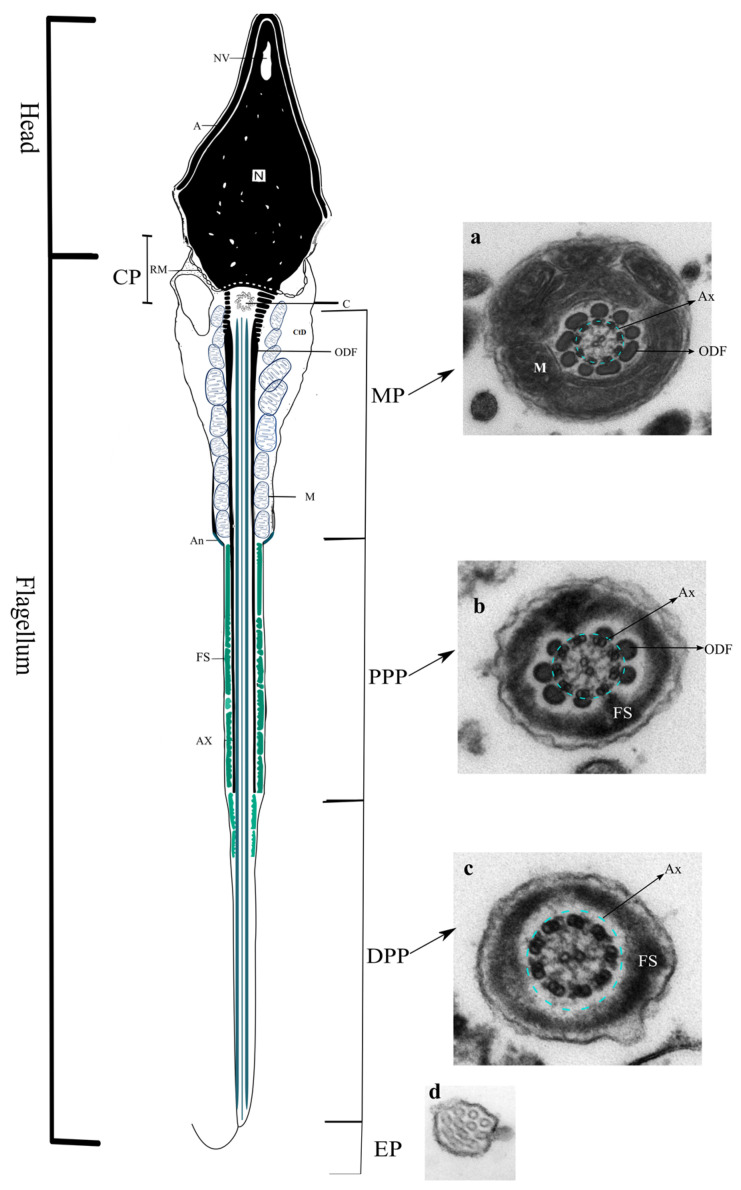
Schematic view of a human sperm cell and transmission electron microscope (TEM) images of a cross-section from the subregions of the sperm flagellum, which is subdivided into the connecting piece (CP), the midpiece (MP), the principal piece (PP; which can be further subdivided into proximal PP (PPP) and distal PP (DPP)), and the endpiece (EP). (**a**) Cross-section of the sperm MP showing the axoneme (Ax, light-blue dotted circle), the outer dense fibers (ODFs), and the mitochondrial sheath (M). (**b**) Cross-section of the sperm PPP showing the axoneme (Ax, light-blue dotted circle), the ODF, and the fibrous sheath (FS). (**c**) Cross-section of the sperm DPP showing the axoneme (Ax, light-blue dotted circle), and the FS. (**d**) Cross-section of the sperm EP, showing the axonemal disorganization that distinguishes this region. Legend: A, acrosome; An, annulus; C, proximal centriole; CtD, cytoplasmatic droplet; N, nucleus; NV, nuclear vacuole; RM, redundant membrane.

**Figure 3 genes-14-00383-f003:**
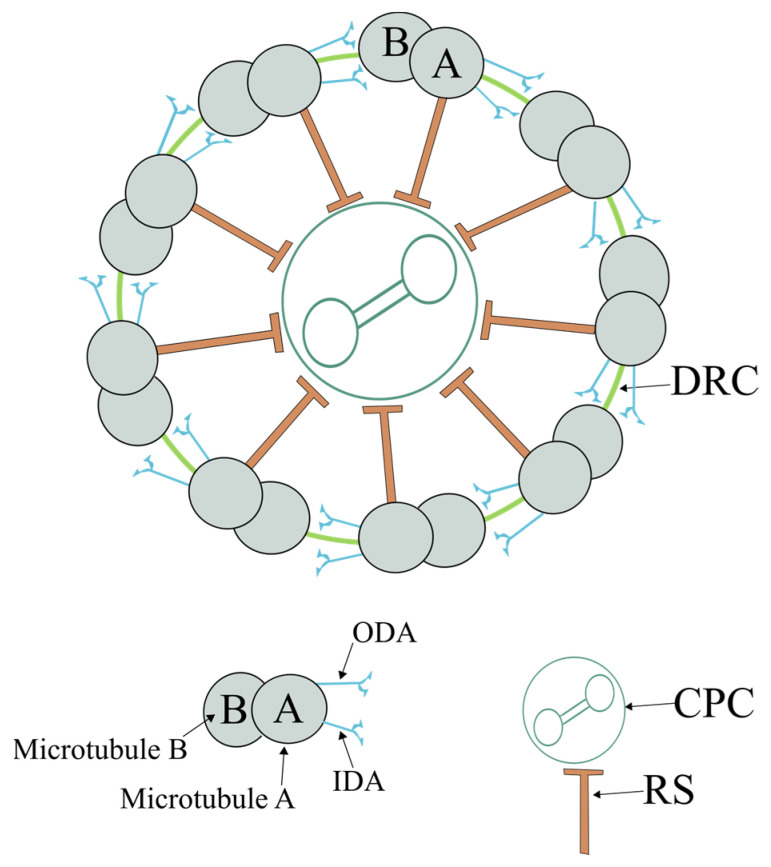
Schematic view of an axoneme (Ax). The Ax is composed of nine axonemal doublet microtubules (also known as the peripheral doublets) linked by the dynein regulatory complex (DRC) and connected by radial spokes (RSs) to a single pair of central Mt, which are surrounded by a fibrillar central sheath, constituting the central pair complex (CPC). The peripheral doublets are composed of microtubule A and microtubule B. From each microtubule A arise two dynein arms: the outer (ODA) and inner (IDA) dynein arms.

**Table 1 genes-14-00383-t001:** Genes in which mutations or gene expression changes were identified with the respective infertility phenotype associated.

Infertility Phenotype	Genes
Asthenospermia	*AKAP3*, *AKAP4*, *AXDND1*, *CATSPER1*, *CATSPER2*, *CATSPER3*, *CATSPER4*, *CCDC103*, *CCDC40*, *CFAP43*, *CFAP44*, *CFAP70*, *COPS7A*, *CRHR1*, *CUL3*, *DEFB126*, *DNAAF1*, *DNAAF6*, *DNAH6*, *DNAH11*, *DNAH17*, *DNAH5*, *DNAH8*, *DNAH9*, *DNAI1*, *DNAJB13*, *DNHD1*, *DRC1*, *HIP1*, *HTX11*, *INSL6 IQCG*, *IQUB*, *KLHL7*, *KRT34*, *LRRC6*, *MT-C03*, *NEDD4*, *NSUN7*, *QRICH2*, *RSPH3*, *RSPH6A*, *SEPTIN4*, *SLC26A8*, *SPAG17*, *SPATA33*, *TEKT2*, *ZMYND10*
Multiple morphological anomalies of the flagella (MMAF)	*BRWD1*, *CCDC34*, *CCDC39*, *CEP135*, *CFAP251*, *CFAP58*, *CFAP61*, *CFAP69*, *CFAP74*, *DNAH1*, *DNAH10*, *DNAH17*, *DNAH2*, *DNAH5*, *DNAH6*, *DNAH7*, *DNAH8*, *DZIP1*, *DZP1*, *FSIP2*, *MAATS1*, *ODF2*, *QRICH2*, *SPAG6*, *SPATA16*, *SPEF2*, *TTC21A*, *TTC29*, *WDR19*, *WDR66*
Nonobstructive azoospermia	*AR*, *ABLIM1*, *AHRR*, *ART3*, *ATM*, *AZFa*, *AZFb*, *AZFc*, *BCL2*, *BPDY2*, *BPY2*, *CCDC34*, *CDC42BPA*, *CDY2A*, *DAZ1*, *DBX3Y*, *DMC1*, *DMRT1*, *DNMT3B*, *EPSTI1*, *ETV5*, *FANCM*, *GNAO1*, *HLA-DRA*, *HSF2*, *HSFY1*, *KLHL10*, *M1AP*, *MCM8*, *MEIOB*, *MLH3*, *MSMB*, *MTHFR*, *NANOS1*, *NPAS2*, *NR5A1*, *PACRG*, *PIWIL2*, *PNLDC1*, *PYGO2*, *RBMX*, *RBYMIAI*, *REC8*, *SIRPG*, *SOHLH1*, *SOX5*, *SPINK2*, *SRSF6*, *STAG3*, *STX2*, *SYCE1*, *SYCE1L*, *SYCP3*, *TAF4B*, *TDRD9*, *TEX11*, *TEX14*, *TEX15*, *USP9Y*, *WT1*, *XRCC2*, *ZMYND15*
Obstructive azoospermia	*ADGRG2*, *CFTR*
Oligozoospermia	*AXDND1*, *DAZ1*, *DAZ2*, *DICER1*, *DNMT1*, *EPHX2*, *GSTM1*, *GSTT1*, *KIT*, *KITLG*, *NR0B1*, *NR5A1*, *OR2W3*, *PARP1*, *PIWIL3*, *PIWIL4*, *PLK4*, *PON1*, *PON2*, *PRM1*, *PSAT1*, *SIRPA*, *SOX6*, *USP8*, *ZMYND15*
Teratozoospermia	*AURKC*, *BSCL2*, *CCIN*, *CCDC90B*, *CCDC91*, *DPY19L2*, *SPATA20*, *SPA17*, *CYP1A1*, *FBXO43*, *PPP2R3C*, *SEPTIN12*, *ZPBP*, *DPY19L2*, *PICK1*, *SPATA16*, *SEPTIN4*

The following databases on phenotype–genotype associations were used as sources of information to elaborate this table: (1) OMIM (https://www.omim.org/); (2) ClinVar (https://www.ncbi.nlm.nih.gov/clinvar/); (3) Human Gene Mutation Database (HGMD, https://www.hgmd.cf.ac.uk/ac/index.php); (4) Human Phenotype Ontology (https://hpo.jax.org/app/about). All databases were accessed on 4 January 2023.

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
