# Peer review of "Morphological and Molecular Bases of Male Infertility: A Closer Look at Sperm Flagellum"

_genes, 2023, doi:10.3390/genes14020383_

Round 1

Reviewer 1 Report

The manuscript entitled “Morphological and Molecular Bases of Male Infertility: a closer look to sperm flagellum” gived a brief overview of the typical sperm flagellum morphology and compile some of the most relevant information regarding the genetic factors with a focus on sperm immotility and on genes related to sperm flagellum development, structure or function. The results are promising. However, some revisions are necessary. 1. There are so many genes involved in the manuscript that it would be easier to understand if the authors could summarize them into diagrams. 2. The author has made a detailed introduction to the genes related to spermatogenesis and flagella, but can the authors conclude which genes are clearly related to asthenospermia or male infertility? 3. Were there any treatments for the flagellate-related sterility disease described in the manuscript? The author can give some prospects for the research

Author Response

The manuscript entitled “Morphological and Molecular Bases of Male Infertility: a closer look to sperm flagellum” gived a brief overview of the typical sperm flagellum morphology and compile some of the most relevant information regarding the genetic factors with a focus on sperm immotility and on genes related to sperm flagellum development, structure or function. The results are promising. However, some revisions are necessary.

  1. There are so many genes involved in the manuscript that it would be easier to understand if the authors could summarize them into diagrams.

A: Dear review, thank you for your comments. A new image (figure 1) including a summary of the main proteins associated to the principal structures from the sperm flagellum was included.

  1. The author has made a detailed introduction to the genes related to spermatogenesis and flagella, but can the authors conclude which genes are clearly related to asthenospermia or male infertility?

A: Dear review, all of the referred genes could potentially lead to infertliliy. But, indeed, not all were specifically associated to infertility. To facilitate the readers understanding, we include a table (table 1) that summarize the main genes in which mutations or gene-expression changes were identified the respective infertility phenotype associated.

  1. Were there any treatments for the flagellate-related sterility disease described in the manuscript? The author can give some prospects for the research

A: In the last section we discuss about the existence of an effective therapy for sperm immotility. Nevertheless, we include this sentence to complement

“So far, there is no effective therapy for sperm immotility, thus, to conceive a child couples need to use Assisted Reproductive Techniques (ART), particularly by intracytoplasmic sperm injection (ICSI). Although there are some successful ICSI performed with immotile spermatozoa reported, the fertilization rate after ICSI with immotile spermatozoa is usually low [220]. The success of the ICSI, were proposed to be related with individuals genotype[221]  Besides, sperm immotility is mostly caused by defects in multiple genes and/or by the interaction of multiple variants located in different genes, thus using ART has the risk of transmitting the mutation to its offspring.”

Reviewer 2 Report

Dear Editor,

The paper of Rute Pereira and Mário Sousa “Morphological and Molecular Bases of Male Infertility: a closer  look to sperm flagellum” gives a detailed overview of various factors affecting the s of male infertility. The review is written in a good language, a lot of important literary data is given. I have a few comments that, in my opinion, can complement the content of the review. Please make the changes noted below.

Although I wrote that the article needs major revision, I am sure that the authors are able to quickly supplement the review of the necessary information and correct of inaccuracies.

 Major points

It is necessary to consider in more detail the transformation during spermiogenesis of the centriolar apparatus. The transition from the centrosome morphology typical for somatic cells, first in the early spermatid, to the origin of the flagellum on the more young daughter dystal centriole and the formation on the more mature maternal centriole of a structure specific only for spermatids - the centriolar adjunct (Fawcett, Phillips,1969; Fawcett, 1981), which is similar in morphology to the primary cilium of somatic cells (absence of central microtubules , lack of motility, probable sensory function, formula (2.5 x 9 + 0) It has been shown that normally the centriolar adjunct disappears in most spermatozoa of healthy donors and in all spermatozoa of boars-producers.In spermatozoa and males with idiopathic male sterility, the centriolar adjunct, on the contrary, remains , which served as the basis for the assumption that this anomaly is one of the causes of idiopathic male infertility (Garanina et al. 2019).

Fawcett, D.W.; Phillips, D.M. The fine structure and development of the neck region of the mammalian spermatozoon. Anat. Rec. 1969, 165, 153–164.

Fawcett, D.W. The Cell. Second edition. Chapter 12: Centrioles; W.B. Saunders Company Press: Philadelphia, London, Toronto, Mexico City, Rio de Janeiro, Sydney, Tokyo, 1981; pp. 551–574, ISBN 0-7216-3584-9.

Garanina et al., (2019) The Centriolar Adjunct–Appearance and Disassembly in Spermiogenesis and the Potential Impact on Fertility. Cells, 8(2), 180; https://doi.org/10.3390/cells8020180.

Lines 239-241

“How CP microtubules assemble is not clear, but studies have suggested that CP microtubules self-assemble without requiring a template within the cilium/flagellum…”

In motile cilia of the ciliary epithelium, central microtubules form on the surface of the basal plate, which delimits the internal cavity of the transition zone (where the walls of the axoneme already contain microtubule doublets, but there are no central microtubules yet) and the cavity of the main zone of cilia (Alieva et al., 2018). Apparently, there is still a specific nucleation center in this area, since an accumulation of electron-dense material is detected morphologically by transmission electron microscopy at the ends of the central microtubules in the central region of the basal lamina.

Alieva I., Staub C., Uzbekova S., Uzbekov R. (2018) A question of flagella origin for spermatids – mother or daughter centriole? In Flagella and Cilia: Types, Structure and Functions. (Editor ed. R. Uzbekov), Chapter 5. New York, USA: Nova Science Publishers, Inc. pp. 106-126.  

Minor points

Line 145

«forming the 9+2 microtubules pattern (Figure 2)»

More correct formula (9x2+2), since it accurately shows the morphology of the flagellum axoneme and the number of microtubules in it is 20, not 11.

Line 148

« composed of 10-protofilaments [43,44] »

It can be understood from the text that microtubule "B" is smaller than microtubule "A" in diameter, since it consists of a smaller number of protofilaments, but this is clearly not the case. It should be added that 4 protofilaments are common for microtubules A and B. Thus, a microtubule "B" has a total of 14 protofilaments and has a slightly larger diameter than microtubule "A".

The cavity of the basal body (centrioles) is delimited from the cavity of the transition zone by the terminal plate.

Line 258

Here it is necessary to indicate that the distal centriole in early (round) spermatide with its proximal end is directed to the surface  of the proximal centriole (this centriole have both "free" ends), that is, the distal centriole is a daughter centriole of mother proximal centriole in the diplosome (this occurs from the formation of the procentriole on the surface of the mother centriole in the last cell cycle). It should also be noted here that a centriolar adjunct appears on the proximal centriole in the spermatid, and later normally disappears in a mature spermatozoon (Garanina et al., 2019).

Line 315

Further, you write about this in more detail, but here it is also necessary to point out that the mitochondrial helix does not consist of one mitochondrion, but is a complex of several dozen individual mitochondria interconnected.

Line 335

“typical (??) morphology “

Why question ? Typical or not?

Line 392

“major part of the sperm flagellum”

major part of the human sperm flagellum

In other species, the length of the flagellum is different.

Line 471

“(with the 9d+2s structure)”

(with the 9x2+2 structure) is more correct.

Line 563

Many proteins in seminal and oviduct fluids exist in membranous vesicles – smaller exosomes and larger extracellular vesicles. (Alninana et al., 2017; Candenas,  Chianese, 2020).  

Alninana et al., (2017) Oviduct extracellular vesicles protein content and their role during oviduct-embryo cross-talk. Reproduction, 154 (3) REP-17-0054 DOI: 10.1530/REP-17-0054

Candenas L. and  Chianese R. (2020) Exosome Composition and Seminal Plasma Proteome: A Promising Source of Biomarkers of Male Infertility. Int J Mol Sci. ; 21(19): 7022. doi: 10.3390/ijms21197022

Extra comments

There are a lot of abbreviations in the text. In order to make it easier for an unprepared reader to understand the content of the article (especially when the decoding of the abbreviation is given in one of the previous sections of the article), I would recommend that the authors to put all the abbreviations in a special section at the end of the review.

intraflagellar transport (IFT)

oligoasthenoteratozoospermia,  OAT

non-coding RNAs (ncRNA)

intramanchette transport (IMT)

protein required for CP assembly

connecting piece (CP),

midpiece (MP),

principal piece (PP), which can be subdivided into

proximal PP (PPP)

distal  PP (DPP)

the end piece (EP).

outer dense fibers (ODF)

fibrous sheath (FS)

central pair complex (CPC),

dynein arms (DA)

inner dynein arms (IDA)

outer dynein arms (ODA)

dynein regulatory complex (DRC)

two central microtubules (C1 and C2, also named as central pair (CP)

radial spokes (RS)

ciliary beat frequency (CBF)

distal centriole (DC)

proximal centriole (PC)

tricarboxylic acid (TCA)

reactive oxygen species (ROS)

succinate dehydrogenase (SDH)

Also, there are many word wraps in the text from one line to another. For ease of reading, especially for non-English readers, I would recommend authors to reduce the amount of word hyphenation.

Author Response

The paper of Rute Pereira and Mário Sousa “Morphological and Molecular Bases of Male Infertility: a closer  look to sperm flagellum” gives a detailed overview of various factors affecting the s of male infertility. The review is written in a good language, a lot of important literary data is given. I have a few comments that, in my opinion, can complement the content of the review. Please make the changes noted below.

Although I wrote that the article needs major revision, I am sure that the authors are able to quickly supplement the review of the necessary information and correct of inaccuracies.

 Major points

It is necessary to consider in more detail the transformation during spermiogenesis of the centriolar apparatus. The transition from the centrosome morphology typical for somatic cells, first in the early spermatid, to the origin of the flagellum on the more young daughter dystal centriole and the formation on the more mature maternal centriole of a structure specific only for spermatids - the centriolar adjunct (Fawcett, Phillips,1969; Fawcett, 1981), which is similar in morphology to the primary cilium of somatic cells (absence of central microtubules , lack of motility, probable sensory function, formula (2.5 x 9 + 0) It has been shown that normally the centriolar adjunct disappears in most spermatozoa of healthy donors and in all spermatozoa of boars-producers. In spermatozoa and males with idiopathic male sterility, the centriolar adjunct, on the contrary, remains , which served as the basis for the assumption that this anomaly is one of the causes of idiopathic male infertility (Garanina et al. 2019).

Fawcett, D.W.; Phillips, D.M. The fine structure and development of the neck region of the mammalian spermatozoon. Anat. Rec. 1969, 165, 153–164.

Fawcett, D.W. The Cell. Second edition. Chapter 12: Centrioles; W.B. Saunders Company Press: Philadelphia, London, Toronto, Mexico City, Rio de Janeiro, Sydney, Tokyo, 1981; pp. 551–574, ISBN 0-7216-3584-9.

Garanina et al., (2019) The Centriolar Adjunct–Appearance and Disassembly in Spermiogenesis and the Potential Impact on Fertility. Cells8(2), 180; https://doi.org/10.3390/cells8020180.

A: Thank you. As you suggest we include this paragraph to further explore the centriole adjunt: “After the formation of the flagellum in the early spermatid, the PC increases the length by addition of material to its free distal end, and the centriolar elongation continues throughout the differentiation of the connecting piece. Ultrastructural studies have observed that the fine structure of the material added to the distal end of the centriole in elongation is distinct to that of the original centriole. Hence, this structure was named as centriolar adjunct to denote that it is not simply a prolongation of the juxtanuclear centriole but is a new structure of slightly different internal organization[94]. The centriolar adjunct is a dynamic structure, that decreases during spermatozoa maturation, and disappearing partially or completely in the mature spermatozoa of different organisms. In humans, it was shown that the disassembly of the centriolar adjunct is required for the functional maturity of human spermatozoa, and an incomplete disassembly of the centriolar adjunct, is associated to infertility [95].”

Lines 239-241

“How CP microtubules assemble is not clear, but studies have suggested that CP microtubules self-assemble without requiring a template within the cilium/flagellum…”

In motile cilia of the ciliary epithelium, central microtubules form on the surface of the basal plate, which delimits the internal cavity of the transition zone (where the walls of the axoneme already contain microtubule doublets, but there are no central microtubules yet) and the cavity of the main zone of cilia (Alieva et al., 2018). Apparently, there is still a specific nucleation center in this area, since an accumulation of electron-dense material is detected morphologically by transmission electron microscopy at the ends of the central microtubules in the central region of the basal lamina.

Alieva I., Staub C., Uzbekova S., Uzbekov R. (2018) A question of flagella origin for spermatids – mother or daughter centriole? In Flagella and Cilia: Types, Structure and Functions. (Editor ed. R. Uzbekov), Chapter 5. New York, USA: Nova Science Publishers, Inc. pp. 106-126.  

A: Unfortunately I could not have access to this reference that reviewer suggest to read and elaborate a sentence on this issue, nevertheless we have complemented this section as follows:

“How CP microtubules assemble is not clear, but studies have suggested that CP microtubules self-assemble without requiring a template within the cilium/flagellum and that the tubulin used for CP assembly is imported directly from the cytoplasm [86]. A study from Nakazawa and co-workers, corroborates the previous findings that there is no organizing center for the CP assembly and showed that the space size within the axoneme is an important factor that directs CP formation in the axoneme. The authors suggest that when the CP precursors are present in the axoneme, the CP may form spontaneously without interacting with any template or RSs. In such a case, only the available space and the amount of the precursors may limit the number of the CPs produced [87].”

Minor points

Line 145

«forming the 9+2 microtubules pattern (Figure 2)»

More correct formula (9x2+2), since it accurately shows the morphology of the flagellum axoneme and the number of microtubules in it is 20, not 11.

A: Each microtubule doublet consists of an internal complete microtubule-A, onto which is attached a second external and incomplete microtubule-B, hence, we have doubt if we could do the association that reviewer suggest. Moreover, the term 9+2 pattern is widely accept in the literature, thus, to maintain the literature consistency and do not create confusion in readers we will maintain this nomenclature. Nevertheless, we will reinforce the idea that the 9 refer to the doublets and the 2 refers to central pair microtubules.

Line 148

« composed of 10-protofilaments [43,44] »

It can be understood from the text that microtubule "B" is smaller than microtubule "A" in diameter, since it consists of a smaller number of protofilaments, but this is clearly not the case. It should be added that 4 protofilaments are common for microtubules A and B. Thus, a microtubule "B" has a total of 14 protofilaments and has a slightly larger diameter than microtubule "A".

A: Dear review, the literature is not consensual in this point. According to Inaba et al (doi:10.1126/sciadv.abq381) authors state that “Cryo-EM observation showed a characteristic feature of the doublet as characterized by thinner B-tubule diameter compared with the A-tubule (the B-tubule was responsible for 36.4% of the diameter of the doublet”

The cavity of the basal body (centrioles) is delimited from the cavity of the transition zone by the terminal plate.

Line 258

Here it is necessary to indicate that the distal centriole in early (round) spermatide with its proximal end is directed to the surface of the proximal centriole (this centriole have both "free" ends), that is, the distal centriole is a daughter centriole of mother proximal centriole in the diplosome (this occurs from the formation of the procentriole on the surface of the mother centriole in the last cell cycle). It should also be noted here that a centriolar adjunct appears on the proximal centriole in the spermatid, and later normally disappears in a mature spermatozoon (Garanina et al., 2019).

A: We have include a statement including this information:

“After the formation of the flagellum in the early spermatid, the PC increases the length by addition of material to its free distal end, and the centriolar elongation continues throughout the differentiation of the connecting piece. Ultrastructural studies have observed that the fine structure of the material added to the distal end of the centriole in elongation is distinct to that of the original centriole. Hence, this structure was named as centriolar adjunct to denote that it is not simply a prolongation of the juxtanuclear centriole but is a new structure of slightly different internal organization[94]. The centriolar adjunct is a dynamic structure, that decreases during spermatozoa maturation, and disappearing partially or completely in the mature spermatozoa of different organisms. In humans, it was shown that the disassembly of the centriolar adjunct is required for the functional maturity of human spermatozoa, and an incomplete disassembly of the centriolar adjunct, is associated to infertility [95].”

Line 315

Further, you write about this in more detail, but here it is also necessary to point out that the mitochondrial helix does not consist of one mitochondrion, but is a complex of several dozen individual mitochondria interconnected.

A: We have included: “The human sperm mitochondria are positioned in four helices, and each helix is composed of 11-15 gyres with on average two mitochondria to each gyre [92].”

Line 335

“typical (??) morphology “

Why question ? Typical or not?

A: We are sorry. It was a mistake and was corrected.

Line 392

“major part of the sperm flagellum”

major part of the human sperm flagellum

In other species, the length of the flagellum is different.

A: Thank you. It was corrected.

Line 471

“(with the 9d+2s structure)”

(with the 9x2+2 structure) is more correct.

A: If reviewer does not mind, we will maintain the term 9+2. The term 9+2 pattern is widely accept in the literature, thus, to maintain the literature consistency and do not create confusion in readers we will maintain this nomenclature.

Line 563

Many proteins in seminal and oviduct fluids exist in membranous vesicles – smaller exosomes and larger extracellular vesicles. (Alninana et al., 2017; Candenas,  Chianese, 2020).  

Alninana et al., (2017) Oviduct extracellular vesicles protein content and their role during oviduct-embryo cross-talk. Reproduction, 154 (3) REP-17-0054 DOI: 10.1530/REP-17-0054

Candenas L. and  Chianese R. (2020) Exosome Composition and Seminal Plasma Proteome: A Promising Source of Biomarkers of Male Infertility. Int J Mol Sci. ; 21(19): 7022. doi: 10.3390/ijms21197022

A: Dear reviewer, we are aware that we have missed several genes and proteins that are associated to fertility. We have focus on the sperm flagellum, and given the complexity of sperm proteome is probable that we have missed some sperm flagellum protein. Thus we do not explore seminal plasma proteins neither sperm head proteins. Nevertheless, this information is relevant and we have included briefly this information. “ In the female reproductive tract, spermatozoa are exposed to a complex environ-ment, where molecular signals promote different mechanisms that regulate sperm motility crucial to uterine migration from the cervix to the uterus and fertilization. To travel through the female l tract spermatozoa requires the assistance of the seminal plasma, which are produced by different male sex organs (including testis, the epidi-dymis, the prostate, the Cowper (bulbourethral) and Littre’s (periurethral) glands and the ampulla of the ductus deferens). Seminal plasma is a very rich fluid that contains a plethora of sugars, oligosaccharides, glycans, lipids, proteins, inorganic ions (calcium, magnesium, potassium, sodium and zinc) and small metabolites. Changes in the sem-inal plasma composition were already associated to infertility[10,199-206]. Moreover, seminal exosomes have been identified in human seminal plasma. Exosomes are membranous nanovesicles of endocytic origin that carry and transfer regulatory bioac-tive molecules and mediate intercellular communication between cells and tissues[207]. Research on seminal exosomes have increased throughout the years and several evi-dence have been given that associated seminal exosomes to infertility [208,209]

An example of the relevance of seminal plasma is given by the Semenogelin (SEMG1) protein. As stated above, spermatozoa only become motile after ejaculation. SEMG1 is a seminal plasma protein, that inhibits sperm progressive motility and, im-mediately after ejaculation, it is hydrolyzed by the prostate-specific antigen (PSA), thus removing SEMG1 from the sperm surface. This allows sperm to become motile in the vaginal canal and capacitation to proceed during uterine migration [7,210].”

Extra comments

There are a lot of abbreviations in the text. In order to make it easier for an unprepared reader to understand the content of the article (especially when the decoding of the abbreviation is given in one of the previous sections of the article), I would recommend that the authors to put all the abbreviations in a special section at the end of the review.

intraflagellar transport (IFT)

oligoasthenoteratozoospermia,  OAT

non-coding RNAs (ncRNA)

intramanchette transport (IMT)

protein required for CP assembly

connecting piece (CP),

midpiece (MP),

principal piece (PP), which can be subdivided into

proximal PP (PPP)

distal  PP (DPP)

the end piece (EP).

outer dense fibers (ODF)

fibrous sheath (FS)

central pair complex (CPC),

dynein arms (DA)

inner dynein arms (IDA)

outer dynein arms (ODA)

dynein regulatory complex (DRC)

two central microtubules (C1 and C2, also named as central pair (CP)

radial spokes (RS)

ciliary beat frequency (CBF)

distal centriole (DC)

proximal centriole (PC)

tricarboxylic acid (TCA)

reactive oxygen species (ROS)

succinate dehydrogenase (SDH)

Also, there are many word wraps in the text from one line to another. For ease of reading, especially for non-English readers, I would recommend authors to reduce the amount of word hyphenation.

A: Thank you. We have included a abbreviature list and revised the text.

,

Round 2

Reviewer 2 Report

Dear Editor!

The authors took into account some of the comments, but there are shortcomings that do not yet allow me to recommend the paper in this form for publication.

I will not argue with the authors about writing the axoneme formula. I will only note that in the literature there are both options (9 + 2) and (9x2 + 2). The first variant, as is often the case, has been preserved since the time when the resolution of the microscope and the preparation of the samples did not allow one to see that peripheral MTs form doublets. At present, this variant continues to be used, but it is obviously obsolete, since it incorrectly shows the number and structure of MTs in the axoneme. How can you explain to the reader that the number "9" in the formula stands for 18 microtubules arranged in doublets, and the number "2" stands for two individual microtubules? Isn't it easier to just write 9x2 + 2 instead of explaining what you have given in the text? Then the difference is immediately clear.

Let the authors themselves choose between the more common and the more correct.

I will give another similar example of the massive use of an outdated concept - the use of the terminology of the number of chromosomes not in terms of the amount of DNA, but in terms of the number of visible chromosomes in mitosis (although they are not visible in the interphase nucleus!). That's what they continue to teach in school today! At present (from Howard and Pelc, 1953) it is well known that DNA duplication, and hence the duplication of number of chromosomes, occurs in the S phase of the cell cycle, long before chromatid segregation can be observed in mitosis. And the cell before mitosis is already tetraploid in terms of the amount of DNA (4n). But the old terminology continues to be used by many authors, especially by embryologists. After all, it was previously believed that doubling of chromosomes occurs in mitosis.

As for the comparative size of microtubules A and B with peripheral doublets of cilia, flagella and centrioles, I categorically cannot miss an paper with such a statement in publication.

In addition to the fact that I myself see these microtubules almost daily in my research with an electron microscope, there are many thousands of experimental papers and reviews on this subject. Protofilaments are most clearly shown in works using tannic acid. You can see more details in the works of Professor Dallai, for example, Figure 4b in the article https://www.ncbi.nlm.nih.gov/pmc/articles/PMC7140630/

Maria Giovanna Riparbelli,1 Veronica Persico,1 Romano Dallai,1 and Giuliano Callaini1,2,* Centrioles and Ciliary Structures during Male Gametogenesis in Hexapoda: Discovery of New Models. Cells. March 2020; 9(3): 744. Published online 2020 Mar 18. doi: 10.3390/cells9030744 PMCID: PMC7140630 PMID: 32197383

I didn't find the link you provided in PubMed Medline. Neither by the author name nor by doi.

doi:10.1126/sciadv.abq381

"Your search was processed without automatic term mapping because it retrieved zero results.

But I understand why this quote misled you. Microtubule B is undoubtedly larger in outer diameter than microtubule A, since it contains in total  more protofilaments. But those 4 protofilaments that it has in common with microtubule A protrude into the lumen of A microtubule. Therefore, if we measure the distance from the outer edge (common with MT B) of microtubule A to the outer edge of microtubule B, then it will certainly be less than the total diameter of microtubule A, that is, less than 50 percent of the total size of the doublet. But if we measure the diameter of microtubule B between the innermost protofilament in the cilium and the outermost one, this diameter will be greater than the diameter of microtubule A.

So, in connection with the above, I insist on indicating in the article the fact that, as part of a peripheral doublet, "microtubule B has 4 common protofilaments with microtubule A and is somewhat larger in diameter."

In addition, you need to redo the diagram in Figure 3. The inner ends of the radial spokes must be closely to the central sheath, and do not hang in the air. The outer and inner dynein arms should reach the adjacent microtubule B in the diagram. Otherwise, how does a cilium or flagellum move? Nexin links are also not shown. Microtubule A of the upper doublet in the figure 3 is displaced inside the axoneme. The shape of the two upper microtubules B differs from the shape of other microtubules B in the diagram. The radial spokes extend from microtubules A, and not from the gap between microtubules A and B, as in your diagram.

The diameter of the axoneme is approximately 200 nm, the diameter of the microtubule "A" is 25 nm. Accordingly - the difference is 8 times. In your diagram, this difference is 12.5 times. Why do it wrong when you can do it right?

Please be careful when preparing the figure. The journal makes high demands on the preparation of illustrations.

By the way, unlike the text, the diagram correctly shows the ratio of the diameters of microtubules A and B.

I can recommend you to contact the Editor of the book "Flagella and Cilia" in the ResearchGate network. Ask him to send you this chapter and he will send you the whole book.

 https://www.researchgate.net/publication/327646211_Chapter_5_A_QUESTION_OF_FLAGELLA_ORIGIN_FOR_SPERMATIDS_-_MOTHER_OR_DAUGHTER_CENTRIOLE

On the cover of this book you will find a more accurate and detailed representation of the structure of the axoneme.

Author Response

The authors took into account some of the comments, but there are shortcomings that do not yet allow me to recommend the paper in this form for publication.

I will not argue with the authors about writing the axoneme formula. I will only note that in the literature there are both options (9 + 2) and (9x2 + 2). The first variant, as is often the case, has been preserved since the time when the resolution of the microscope and the preparation of the samples did not allow one to see that peripheral MTs form doublets. At present, this variant continues to be used, but it is obviously obsolete, since it incorrectly shows the number and structure of MTs in the axoneme. How can you explain to the reader that the number "9" in the formula stands for 18 microtubules arranged in doublets, and the number "2" stands for two individual microtubules? Isn't it easier to just write 9x2 + 2 instead of explaining what you have given in the text? Then the difference is immediately clear.

Let the authors themselves choose between the more common and the more correct.

I will give another similar example of the massive use of an outdated concept - the use of the terminology of the number of chromosomes not in terms of the amount of DNA, but in terms of the number of visible chromosomes in mitosis (although they are not visible in the interphase nucleus!). That's what they continue to teach in school today! At present (from Howard and Pelc, 1953) it is well known that DNA duplication, and hence the duplication of number of chromosomes, occurs in the S phase of the cell cycle, long before chromatid segregation can be observed in mitosis. And the cell before mitosis is already tetraploid in terms of the amount of DNA (4n). But the old terminology continues to be used by many authors, especially by embryologists. After all, it was previously believed that doubling of chromosomes occurs in mitosis.

As for the comparative size of microtubules A and B with peripheral doublets of cilia, flagella and centrioles, I categorically cannot miss an paper with such a statement in publication.

In addition to the fact that I myself see these microtubules almost daily in my research with an electron microscope, there are many thousands of experimental papers and reviews on this subject. Protofilaments are most clearly shown in works using tannic acid. You can see more details in the works of Professor Dallai, for example, Figure 4b in the article https://www.ncbi.nlm.nih.gov/pmc/articles/PMC7140630/

Maria Giovanna Riparbelli,1 Veronica Persico,1 Romano Dallai,1 and Giuliano Callaini1,2,* Centrioles and Ciliary Structures during Male Gametogenesis in Hexapoda: Discovery of New Models. Cells. March 2020; 9(3): 744. Published online 2020 Mar 18. doi: 10.3390/cells9030744 PMCID: PMC7140630 PMID: 32197383

I didn't find the link you provided in PubMed Medline. Neither by the author name nor by doi.

A: Dear Reviewer, we understand your point of view and we are not saying that the reviewer is wrong. Indeed, we have an imprecise formula. We will correct to 9d+2 s. In this way it will be clear to readers that we are referring to 9 microtubules doublets and two singlet microtubules. We prefer this formula because by saying 9 doublets it presupposes that we are referring to a complete microtubule, the A-tubule, with 13 protofilaments, and an incomplete microtubule, the B-tubule, tethered to the A-tubule, by microtubule inner proteins, with 10 protofilaments.

But I understand why this quote misled you. Microtubule B is undoubtedly larger in outer diameter than microtubule A, since it contains in total more protofilaments. But those 4 protofilaments that it has in common with microtubule A protrude into the lumen of A microtubule. Therefore, if we measure the distance from the outer edge (common with MT B) of microtubule A to the outer edge of microtubule B, then it will certainly be less than the total diameter of microtubule A, that is, less than 50 percent of the total size of the doublet. But if we measure the diameter of microtubule B between the innermost protofilament in the cilium and the outermost one, this diameter will be greater than the diameter of microtubule A.

So, in connection with the above, I insist on indicating in the article the fact that, as part of a peripheral doublet, "microtubule B has 4 common protofilaments with microtubule A and is somewhat larger in diameter."

A: Dear reviewer, we are happy to include this information “microtubule B has 4 common protofilaments with microtubule A and is somewhat larger in diameter” if you could indicate original articles that we could cite, because we have searched and do not find references that validate this statement, we just see that on the book that you recommend "Flagella and Cilia"”, but it is a book and not an original article.

The Cyo-TEM data that we have found, corroborate the fact that the doublets comprise α- and β-tubulin heterodimers, formed by a complete A-tubule with 13-protofilaments and an incomplete B-Tubule with 10- protofilaments. The A and B tubule interactions are mediated by microtubule inner proteins. Specifically the tubulin protofilaments B1 (from B-tubule) interacts with the outer surfaces of tubulin of protofilaments A10 and A11 (from A-Tubule); and the protofilament A1 connects with B10 [1-5].

Regarding the size, we have not discussed about the size in our manuscript because we agree that by conventional TEM it seem that B-Tubule is larger than A-Tubule. However, Inaba et al Cryo-EM measures show that B-tubule diameter is thinner when compared with the A-tubule [6].

Nevertheless, if the reviewer could send us original articles affirming the opposite, we are happy to include this discussion also in the manuscript.

In addition, you need to redo the diagram in Figure 3. The inner ends of the radial spokes must be closely to the central sheath, and do not hang in the air. The outer and inner dynein arms should reach the adjacent microtubule B in the diagram. Otherwise, how does a cilium or flagellum move? Nexin links are also not shown. Microtubule A of the upper doublet in the figure 3 is displaced inside the axoneme. The shape of the two upper microtubules B differs from the shape of other microtubules B in the diagram. The radial spokes extend from microtubules A, and not from the gap between microtubules A and B, as in your diagram.

The diameter of the axoneme is approximately 200 nm, the diameter of the microtubule "A" is 25 nm. Accordingly - the difference is 8 times. In your diagram, this difference is 12.5 times. Why do it wrong when you can do it right?

Please be careful when preparing the figure. The journal makes high demands on the preparation of illustrations.

By the way, unlike the text, the diagram correctly shows the ratio of the diameters of microtubules A and B.

A: Dear review, we have modified the figure. We give our best to be as precise as possible. We hope that know the figure was suitable.  

I can recommend you to contact the Editor of the book "Flagella and Cilia" in the ResearchGate network. Ask him to send you this chapter and he will send you the whole book.

https://www.researchgate.net/publication/327646211_Chapter_5_A_QUESTION_OF_FLAGELLA_ORIGIN_FOR_SPERMATIDS_-_MOTHER_OR_DAUGHTER_CENTRIOLE

On the cover of this book you will find a more accurate and detailed representation of the structure of the axoneme.

A:Thank you. We have contacted and already have the book.

  1. Downing KH, Sui H, Structural insights into microtubule doublet interactions in axonemes, Curr. Opin. Struct. Biol. 2007;17(2): 253-9. https://doi.org/10.1016/j.sbi.2007.03.013
  2. Ichikawa M, Liu D, Kastritis PL, Basu K, Hsu TC, et al., Subnanometre-resolution structure of the doublet microtubule reveals new classes of microtubule-associated proteins, Nat. Commun. 2017;8(1): 15035. https://doi.org/10.1038/ncomms15035
  3. Ma M, Stoyanova M, Rademacher G, Dutcher SK, Brown A, et al., Structure of the Decorated Ciliary Doublet Microtubule, Cell 2019;179(4): 909-922.e12. https://doi.org/10.1016/j.cell.2019.09.030
  4. Nicastro D, Fu X, Heuser T, Tso A, Porter ME, et al., Cryo-electron tomography reveals conserved features of doublet microtubules in flagella, Proc Natl Acad Sci U S A 2011;108(42): E845-53. https://doi.org/10.1073/pnas.1106178108
  5. Maheshwari A, Obbineni JM, Bui KH, Shibata K, Toyoshima YY, et al., alpha- and beta-Tubulin Lattice of the Axonemal Microtubule Doublet and Binding Proteins Revealed by Single Particle Cryo-Electron Microscopy and Tomography, Structure 2015;23(9): 1584-1595. https://doi.org/10.1016/j.str.2015.06.017
  6. Inaba H, Sueki Y, Ichikawa M, Kabir AMR, Iwasaki T, et al., Generation of stable microtubule superstructures by binding of peptide-fused tetrameric proteins to inside and outside, Science Advances 2022;8(36): eabq3817. https://doi.org/10.1126/sciadv.abq3817
